# A diffusion of innovations measurement scale for reinvention, relative advantage, compatibility, complexity, trialability and observability

Hannah Overbye-Thompson¤*, Kristy A. Hamilton

University of California Santa Barbara, Santa Barbara, United States of America

¤ Current address: Department of Communication – UC Santa Barbara, 4005 Social Sciences & Media Studies, UC Santa Barbara
* Hoverbye@ucsb.edu

## Abstract

Diffusion of innovations (DOI) theory identifies critical factors that influence technology adoption rates and offers a predictive model for understanding how innovations spread through populations. While DOI theory encompasses six key perceptual characteristics (relative advantage, compatibility, complexity, trialability, observability, and reinvention), most empirical research operationalizes only Rogers' five core attributes, rarely integrating reinvention despite its theoretical importance for understanding post-adoption adaptation. This research develops and validates a comprehensive scale measuring all six DOI characteristics, with particular attention to the reinvention construct. Through three independent samples (n = 2,019), we test the scale's validity within a nomological network, creating an adaptable instrument for studying innovation diffusion that captures the full scope of DOI theory.

## Introduction

Diffusion of innovations (DOI) theory is central to scholarship across various disciplines including Communication, Management, and Information Technology. DOI theory identifies critical factors that influence the rate of technology adoption and offers a predictive model to understand human perceptions that contribute to innovation adoption by different segments of the population. DOI is one of the most prominent media effects theories to date with over 94,000 citations [1]. Through application of DOI theory, scholars are able to hone in on processes and outcomes that influence rates of technological adoption. A strength of this theory is its ability to apply to the adoption of a wide range of innovations, such as hybrid corn seed [2], transportation biking [3], insect consumption [4], and autonomous vehicles [5]. Indeed, a major advantage of DOI theory is its tendency to cultivate focus on the capabilities of a variety of innovations. Nonetheless, as new innovations and variations of old innovations

in any medium, provided the original author and source are credited.

**Data availability statement:** The data for this study is available at Open Science Framework: https://osf.io/7p4gm/?view_only=2168ab9b-71314b8eaf3d2bf254ec95ff.

**Funding:** The author(s) received no specific funding for this work.

**Competing interests:** The authors have declared that no competing interests exist.

arise, we may need to revisit the generality of measures we use to operationalize the perceptions of innovation characteristics that are outlined in DOI theory.

The purpose of this research is to create a measure for assessing perceptions of innovations that contribute to innovation diffusion, including the often-overlooked reinvention construct. Reinvention is critical to DOI theory because it reflects how innovations are adapted and reshaped by adopters during the diffusion process—making the innovation more compatible with the adopters' needs, values, or social infrastructure. While DOI theory has been successfully applied to a variety of innovations (cf., [6]), existing measurement approaches face limitations. Most empirical studies operationalize only a subset of Rogers' five core perceptual attributes – relative advantage, compatibility, complexity, trialability, and observability [6]. These studies consistently exclude reinvention despite its theoretical relevance to post-adoption adaptation behaviors that are particularly common in software contexts. This omission of reinvention marks a significant gap in DOI research, especially in the current era shaped by artificial intelligence. Software innovations, unlike many hardware innovations, are characterized by their malleability and capacity for user-driven customization and adaptation. Users frequently modify software applications, configure settings, and adapt functionality to their specific needs. These behaviors align directly with Rogers' conceptualization of reinvention as "the degree to which an innovation is changed or modified by a user in the process of its adoption and implementation" [2]. The iterative nature of software development, with frequent updates and user feedback cycles, makes reinvention particularly relevant for understanding how software innovations diffuse and evolve within user communities.

Over the past half-century, DOI theory has given rise to several measures assessing the six key perceptual characteristics, or attributes of innovations, that determine the perception and use of a particular innovation: relative advantage, compatibility, complexity, trialability, observability, and reinvention [2]. However, existing scales face two primary limitations. First, most studies measure only a subset of the six theoretical constructs, with reinvention being particularly neglected. Second, existing measures are typically developed and validated for single innovations [3,7–10], forcing researchers who want to measure the DOI framework to piece together elements of different scales to adapt them to the innovation under investigation.

We addressed these limitations by developing and validating a comprehensive 19-item survey instrument that measures all six perceived innovation characteristics proposed by DOI theory. The scale uses a flexible template format where researchers can insert specific innovations and tasks (e.g., "[Innovation] allows me to accomplish tasks such as [task] more efficiently"), making it adaptable across different contexts while maintaining measurement consistency. To our knowledge, this is the first validated measure to assess all six DOI characteristics across contexts using a standardized, adaptable format. Across three studies, we generated items, assessed the factor structure of our comprehensive DOI scale, and evaluated its associations with related psychological and behavioral constructs, consistent with its theoretical network.

## Overview of diffusion of innovations

DOI theory explains how innovations (e.g., technologies, ideas, ways of doing things) are adopted, rejected, or reinvented by a population [2]. These innovations often consist of either software or hardware components, which can diffuse throughout a population at different rates [2]. For example, a software such as a videogame will diffuse at a different rate than the console on which the game is played. DOI theory asserts that how an innovation is perceived influences if and how it is adopted. DOI theorizes six perceptual characteristics, or innovation attributes, that influence innovation adoption: compatibility, complexity, relative advantage, trialability, observability, and reinvention.

## Compatibility

Compatibility is the degree to which an innovation is consistent with existing values, needs, and experiences [2]. An innovation can be more or less compatible with (1) cultural values and believes, (2) previous ideas, or (3) the adoptee's needs for the innovation [2]. Perceptions of compatibility are positively associated with innovation adoption. Prior work has measured compatibility within a variety of contexts, such as smartwatches [11], personal workstations [12], online purchasing [11], and mobile banking [13]. Compatibility is generally measured by asking if an innovation meets the needs of the adoptee, is able to seamlessly integrate into one's life, and fits well with the ways that a person likes to do things.

## Complexity

Complexity is the degree to which an innovation is perceived as being hard or difficult to use or understand [2].Complexity has been measured by assessing how frustrating or difficult a person perceives an innovation [11], which is negatively associated with innovation adoption, and by assessing how easy or simple an innovation is to use [12], which is positively associated with innovation adoption. Either approach is an acceptable conceptual definition of complexity. Researchers sometimes prefer ease of use questions so that the directionality of complexity is aligned with other DOI characteristics [14]. Software may be seen as more complex than hardware as the underlying mechanisms that allow software to function (i.e., bits of code) may be more difficult to understand because they often experience continuous updates, can be highly customizable, and are sometimes hidden by design. Complexity has been measured to predict innovation adoption in a variety of contexts, such as evidence-based practices in healthcare [15], e-readers [16], and e-health tools [14].

## Relative advantage

The extent to which an innovation is perceived as superior to its predecessor or to its alternative is known as relative advantage [2]. An innovation may be perceived as high in relative advantage when a person views the innovation as having higher social status, economic status, efficiency, or benefits than the preceding or alternate innovation [2]. Relative advantage is positively associated with innovation adoption and is commonly measured by comparing an innovation under investigation to named or unnamed prior or current alternatives. For example, participants may be asked if an innovation enables them to generally accomplish tasks more quickly [12,13], or by comparing the innovation to a specific alternative [11]. Hardware innovations often bring tangible improvements in terms of performance, durability, or functionality, making it easier for users to recognize their advantages over older hardware. Software innovations often bring improvements by allowing users to access different affordances or features, which are often rolled out slowly to an already existing software package. Relative advantage has been used to explain the diffusion of innovations such as cover cropping [17] and smartwatches [11].

## Observability

Observability is the degree to which an innovation is noticeable or communicable to others, and is positively associated with innovation adoption. When an innovation is observable, people can see how it benefits others and how to use the innovation. In general, software-based technologies are less observable than hardware-based technologies [2]. Users

can physically see and touch hardware devices, making it easier to assess their quality and functionality. This visibility can lead to quicker adoption as users have a clearer understanding of the innovation's use and advantages. Observability has been measured by asking the degree to which people are able to see the benefits of an innovation (e.g., [12]), how to use the innovation, and how others are using the innovation. Observability has been used to explain innovation adoption in contexts such as adoption of mobile applications [9] and personal work stations [12].

## Trialability

Trialability is the degree to which an individual is able to try or test an innovation before adopting it, and is positively associated with innovation adoption [2]. For example, if someone has a two-week free trial of a videogame, they may be more likely to try the game to see if they would like to adopt it. Trialability is often assessed by asking participants if they were able to use an innovation before adopting it [11,12], or if trialing an innovation contributed to their adoption decision. Hardware innovations can be easier to test compared to software. Users can physically interact with hardware innovations and see how they fit into existing routines. This hands-on experience can lead to quicker adoption, as users can assess the hardware's compatibility and usefulness more directly. Conversely, the barrier to trying certain software may be lower than hardware as one can often trial new software from home. Trialability has been associated with innovation adoption in contexts such as cover crops [17] and e-health [14].

## Reinvention

Reinvention is the degree to which a user [21] changes or modifies an innovation [2]. After adopting an innovation, users often adapt or reinvent the innovation to better suit their needs [18,19]. Reinvention is associated with increased rates of innovation use [2] and greater sustainability or continuity of use. Reinvention has been thought of as both an innovation adoption type (i.e., an innovation can either be adopted, rejected, or reinvented) and also as a perceptual characteristic of an innovation (e.g., one could perceive an innovation to be easily customizable) [20]. Reinvention has been used to explain, among other innovations, how political policies change over time [21] as well as how modifying messages changes the diffusion of information in social networks [22].

Despite both Rogers' conceptualization of reinvention as a core innovation characteristic [2,19] as well as its usefulness in explaining innovation adoption [19,21,22] empirical DOI research has excluded this construct for several reasons. First, there is a historical bias toward pre-adoption factors: early diffusion studies prioritized the five core perceptual characteristics (relative advantage, compatibility, complexity, trialability, and observability) that influence initial adoption decisions [23,24], whereas reinvention often occurs post-adoption, during the implementation phase [19]. As a result, reinvention has been treated as a peripheral outcome rather than a central perception guiding adoption decisions.

Second, prior studies have typically focused on hardware innovations or tightly scoped software contexts, where reinvention was either less observable or difficult to quantify [5,8,9,14,17]. For example, early DOI studies examining the adoption of agricultural equipment or medical devices focused on binary adoption decisions where users had limited opportunities to modify the innovation's core functionality. While this approach worked well for many hardware innovations, software innovations present fundamentally different characteristics that challenge traditional DOI measurement approaches; hardware innovations are often more tangible and observable than software. Users can physically see and touch hardware devices, making it easier to assess their quality and functionality. Hardware innovations also have different life cycles and costs than software. Therefore, they are promoted differently and have distinct competitive dynamics. While hardware may diffuse more rapidly in certain circumstances, software innovations have unique advantages and characteristics that can also lead to rapid adoption. Among software innovations, reinvention is likely to play a central role in determining whether and how an innovation will be adopted and sustained. Unlike hardware, software is inherently malleable as users can adjust configurations, find novel use cases, and engage in iterative customization over time. These adaptation behaviors align directly with the concept of reinvention and help explain post-adoption engagement, variation

in use, and long-term retention. Innovations that afford greater reinvention may be perceived as more useful, flexible, and compatible with individual workflows, which can in turn increase diffusion rates.

Third, and finally, conceptual ambiguity and definitional inconsistency have impeded the inclusion of reinvention in empirical DOI studies. Rogers [2] conceptualized reinvention as both an adoption outcome (innovations can be adopted, rejected, or reinvented) and a perceptual characteristic (some innovations are perceived as more modifiable), creating theoretical confusion among researchers about how the construct should be measured. Additionally, temporal complexity poses barriers to measuring reinvention because reinvention unfolds dynamically throughout adoption and implementation processes, as demonstrated by [19]. This temporal complexity makes it difficult for researchers to capture ongoing modifications and establish appropriate measurement timeframes. Context dependency further complicates measurement as reinvention manifests differently across organizational and technological contexts, creating trade-offs between context-specific validity and theoretical generalizability [25].

This paper resolves conceptual ambiguity by explicitly treating reinvention as a perceptual characteristic (i.e., users' beliefs about an innovation's modifiability) rather than as an adoption outcome. By focusing on reinvention as a perception, we can capture individual-level variation in how users anticipate shaping or adapting an innovation, even if they have not yet done so in practice. This is especially useful in digital and software contexts, where modifiability is often expected and innovations are frequently tailored in real time. Defining reinvention as a perceptual characteristic also avoids conflating the causes and consequences of reinvention, allowing researchers to examine its influence on adoption separately from its downstream effects. In addition, treating reinvention as a perceptual characteristic offers practical advantages for measurement. Specifically, it allows researchers to assess reinvention without needing to track post-adoption behaviors, which pose logistical challenges. Many studies lack the resources to follow users over time, and as a result, important dimensions of innovation adoption are overlooked. A perception-based approach provides a feasible alternative that still captures meaningful variation in how users engage with innovations. Finally, we tackle generalizability challenges through a flexible template approach that allows researchers to insert specific innovations and tasks while maintaining measurement consistency across contexts. Incorporating reinvention into DOI models provides explanatory leverage to account for seemingly similar software tools that experience divergent adoption outcomes. By including reinvention as a core measured attribute, this paper extends the DOI framework's applicability to digital innovation ecosystems where user-driven modification is a normative and expected behavior.

Beyond establishing a measure for reinvention, the goal of this research is to develop a scale to measure the well-established constructs within DOI theory. Although compatibility, complexity, relative advantage, trialability, observability, and reinvention have been measured extensively throughout the lifespan of DOI theory, most measures either do not focus on assessing software innovations (e.g., [11,12]), have not been updated in the past decade, or do not measure all six perceptual characteristics of DOI theory [11,12,26]. To address these gaps, this research develops a flexible DOI scale that measures the perceptual characteristics of compatibility, complexity, relative advantage, trialability, observability, and reinvention across varying software contexts. To develop and validate this scale we follow the recommendations of [27] through multi-sample validation within a robust nomological network.

## Study 1: Item adaptation and CFA

Study 1 adapts questions measuring relative advantage, compatibility, complexity, trialability, observability, and reinvention to four different image recognition software contexts: facial recognition for phone unlock, social media filters, facial recognition for financial technology (i.e., biometric security), and image sensing software such as automatic water faucets. We selected these contexts as they are (1) software-based, (2) used across a variety of settings in daily life, and (3) offer different levels of alternatives to adoption. Although all four contexts employ image recognition technology, they represent distinct software innovations that we suspect will vary across perceptual DOI characteristics. While facial recognition software may come pre-installed on devices such as phones, its adoption does not always match the diffusion of the phone's

hardware. For instance, a user might choose to unlock their phone with a pin or a different kind of passcode instead of facial recognition. Similarly, a user could use a password instead of biometric security when accessing their bank account, and using social media does not mandate the use of facial filters; users have the option to select non-facial filters or not to use any filters at all. Conversely, image sensors often diffuse at the same rate as their hardware; a water faucet that uses an image sensor generally does not come with a manual alternative. By measuring four distinct contexts of image recognition technology, we are able to create a measure designed to be adapted to various software innovations.

We seek to build a flexible measure of relative advantage, compatibility, complexity, trialability, observability, and reinvention by creating a set of questions adapted from prior measures of DOI theory, cited below. Study 1 evaluates the factor structure of these items. As each DOI characteristic is conceptually distinct, items should load onto a six-factor CFA model. If the items load onto a six-factor solution consisting of at least three items per factor, then this suggests that the measures are appropriate for measuring the latent DOI perceptual characteristics and that each characteristic is conceptually distinct from one another.

## Methods

### Participants

We recruited 325 participants from an undergraduate research subject pool consisting of students taking classes in a communication department from May to June 2023. While the sample size required for a CFA is debated, we adhered to the recommendation by [28] of a minimum sample size of 265 to accommodate any non-normal factor indicators. Of the sample, 61% identified themselves as women (n = 199), 32% as men (n = 105), 2% as non-binary (n = 5), and 5% did not specify their gender (n = 16). Participants' ages ranged from 19–25, with an average age of 19.95 (SD = 1.64). 22% identified as Asian (East Asian 58, South Asian 12, Mixed Asian 2), 2% as Black/African American (7), 15% as Latino (49), 38% as White (123), 3.6% as mixed White/Latino (12), and 15% as either mixed race or other (49). Thirteen participants (0.4%) did not specify their race.

This research was reviewed and approved by the University of California, Santa Barbara Human Subjects Committee (Protocol #4-23-0221). The study was determined to be exempt under Category 2 of 45 CFR 46.104(d), which covers survey procedures involving adult participants. All participants provided written informed consent prior to participation.

### Procedure

In the initial phase of our scale development process, we drew upon existing DOI measures to construct six subscales measuring relative advantage [13], compatibility [12,13,16], complexity [12], trialability [12,14], observability [12,29], and reinvention [2]. This preliminary set of items was derived from a careful review of DOI literature, with a focus on adapting questions that had demonstrated utility in previous research contexts. Considering the well-developed theoretical foundation of DOI research, we determined that adapting existing measures would be both efficient and would allow for better comparability with prior studies. To evaluate the applicability of these questions, we asked participants to respond to the set of questions across four different image recognition-based software contexts as discussed above: facial recognition for phone unlock (phone unlock), social media filters, facial recognition for financial technology (finances), and image sensing software such as automatic water faucets (image sensors).

We conducted a confirmatory factor analysis (CFA) with maximum likelihood estimation using a variance-covariance matrix on Mplus [30]. Following the recommendation of [27] we chose a CFA over an exploratory factor analysis (EFA) because we (1) adapted existing scales and (2) have a sound theoretical basis and understanding of the underlying factor structures of DOI. We specified and ran four different models, one for each image recognition-based software contexts (i.e., phone unlock, finances, social media filters and image sensors), each of which comprised six factors, one for each of the six diffusion of innovation perceptual characteristics under investigation (i.e., compatibility, complexity, relative advantage, observability, trialability and reinvention).

## Measures

We measured all responses on a 5-point scale from (1) strongly disagree to (5) strongly agree.

### Relative advantage

We used four items adapted from [13] to assess the degree to which the innovation is perceived as being better than preceding innovations. For example, "[innovation] allows (would allow) me to accomplish tasks, such as [task], more efficiently," (e.g., "Facial recognition technology allows (would allow) me to accomplish tasks, such as unlocking my phone, more efficiently") (Phone Unlock, $M=3.64$, $SD=0.66$; Finances, $M=3.70$, $SD=0.89$; Social Media Filters, $M=2.99$, $SD=0.93$; Image Sensors, $M=3.58$, $SD=0.78$).

### Compatibility

Four items adapted from [16] evaluated the perceived compatibility of image recognition-based software innovations with participants' past experiences, beliefs, and values. For example, "[innovation] fits (would fit) well with the way that I like to [task]," (e.g., "Facial recognition technology fits (would fit) well with the way that I like to use my phone.") (Phone Unlock, $M=4.05$, $SD=0.95$; Finances, $M=3.9$, $SD=1.05$; Social Media Filters, $M=3.19$, $SD=1.16$; Image Sensors, $M=3.77$, $SD=0.88$).

### Complexity

We adapted four items from [12] to measure the perceived complexity of software-based innovations. These items assess how participants view the ease of understanding a technology. For example, "It is (would be) easy to get [innovation] to do what I want them to do when using them to [task]," (e.g., "It is (would be) easy to get facial recognition algorithms to do what I want them to do when using them to unlock a phone.") (Phone Unlock, $M=3.91$, $SD=0.72$; Finances, $M=3.87$, $SD=0.77$; Social Media Filters, $M=3.68$, $SD=0.78$; Image Sensors, $M=3.66$, $SD=0.76$).

### Observability

To assess participants' perceived level of observability of software-based innovations, we measured four items related to the degree to which the results of innovations are observable to the participant (adapted from [29]). For example, "I am (would be) able to observe when others in my environment use [innovation] to [task]," (e.g., "I am (would be) able to observe when others in my environment use facial recognition technology to unlock a phone.") (Phone Unlock, $M=3.42$, $SD=0.85$; Finances, $M=2.92$, $SD=1.02$; Social Media Filters, $M=3.51$, $SD=0.95$; Image Sensors, $M=3.40$, $SD=0.85$)

### Trialability

We adapted four items from [14] to measure perceived trialability, which evaluates the ability to use software-based innovations before deciding to adopt them. For example, "I have (anticipate having) the ability to try out [innovation] to accomplish [task] before deciding whether I like it or not," (e.g., "I have (anticipate having) the ability to try out facial recognition technology to unlock a phone before deciding whether I like it or not.") (Phone Unlock, $M=4.06$, $SD=0.75$; Finances, $M=3.70$, $SD=0.97$; Social Media Filters, $M=3.64$, $SD=0.88$; Image Sensors, $M=3.63$, $SD=0.84$).

### Reinvention

We measured four items adapted from [2] to evaluate participants' perceived level of reinvention—the extent to which users can or do change or modify image recognition-based software innovations. For example, "I often have (anticipate having) to experiment with new ways of using [innovation]," (e.g., "I often have (anticipate having) to experiment with new ways of using facial recognition technology when using it to unlock my phone.") (Phone Unlock, $M=2.58$, $SD=0.84$; Finances, $M=2.62$, $SD=0.88$; Social Media Filters, $M=2.77$, $SD=0.85$; Image Sensors, $M=2.85$, $SD=0.81$).

For the full texts of the initial measures see S2 Tables 1–4.

## Results

To evaluate our data, we used standard fit criteria, considering models with a SRMR ≤ .08, a CFI and TLI ≥ .95, and an RMSEA < 0.08 a good fit [31–33].

The results of the CFA suggested a reasonable, although not ideal, fit with values of SRMR exceeding 0.08 and TLI/CFI values falling below .95 across all models; RMSEA for all models met acceptable fit criteria, with the RMSEA falling below .08. See Table 1 for model fit statistics and Table 2 for factor loadings.

To improve the model fit, we removed items with factor loadings less than 0.6. We removed the third relative advantage question for all contexts, e.g., "The disadvantages of using [innovation] to [task] (would) outweigh the advantages"; the fourth complexity measure for all contexts, e.g., "Using [innovation] to [task] is (would be) cumbersome"; the fourth trial-ability measure for all contexts, e.g., "I have not had much opportunity to try [innovation] to [task] in the past"; the fourth reinvention measure for all contexts, e.g., "I rarely have (anticipate having) to come up with novel ways to get [innovation] to work for me when using it to accomplish [task]"; and the first observability measure for phone unlock and image sensor contexts, e.g., "Changes in others' use of [innovation] would be) obvious to me."

The results of the CFA analysis on the remaining items suggest satisfactory models on all goodness of fit statistics, with the SRMR less than 0.08 and TLI/CFI values falling above or equal to .95 across all models. As with the initial CFA, the RMSEA for all models met acceptable fit criteria, with the RMSEA falling below .08. See Table 3 for model fit statistics and Table 4 for factor updated factor loadings. To ensure parsimony we eliminated two observability items from the social media filter and finance contexts so that three questions were associated with observability across contexts.

### Study 1 discussion

In Study 1, we reviewed existing literature to generate questions for measuring six perceptual innovation characteristics that underlie DOI theory. We also collected data to evaluate how the initial set of items perform across different types of software-based innovations. Study 1 offers initial support for a six-factor measure of perceptual innovation characteristics, consisting of relative advantage, compatibility, complexity, trialability, observability, and reinvention. The CFA results showed satisfactory model fit across all four contexts after item refinement, with a SRMR < 0.08, CFI/TLI ≥ 0.95, and RMSEA < 0.08; all items loaded well onto their respective factors (loadings > 0.6) and the six-factor structure was consistent across all four software-based contexts, suggesting adaptability. However, given the item modification, the scale requires further validation. Therefore, we cross-validated our results on an independent sample in Study 2.

### Study 2: Replicating CFA and testing a nomological network

Study 2 employed the same procedure as in Study 1. The goal of this study is to use a new sample to evaluate the updated questions and to test for internal discriminant validity by examining the correlations between factors within each context. Additionally, Study 2 evaluates a nomological network of concepts [34] surrounding DOI.

**Table 1. Fit indices for sample 1 initial CFA.**

| Model | χ² (df), p-value | RMSEA | CFI | TLI | SRMR |
|---|---|---|---|---|---|
| Phone Unlock | 538.271 (df = 237), p < .001 | 0.064 | 0.90 | 0.884 | 0.086 |
| Finances | 586.775 (df = 237), p < .001 | 0.069 | 0.90 | 0.89 | 0.089 |
| Social Media Filters | 560.146 (df = 237), p < .001 | 0.066 | 0.92 | 0.90 | 0.102 |
| Image Sensors | 702.263 (df = 237), p < .001 | 0.080 | 0.85 | 0.83 | 0.103 |

**Table 2. Factor Loadings for Initial CFA.**

| Social Media Filters | | | Image Sensors | | | Finances | | | Phone Unlock | | |
|---|---|---|---|---|---|---|---|---|---|---|---|
| param | est | se | param | est | se | param | est | se | param | est | se |
| RS1 | .893 | .019 | RI1 | .848 | .025 | RF1 | .792 | .035 | RU1 | .785 | .032 |
| RS2 | .88 | .019 | RI2 | .812 | .034 | RF2 | .75 | .032 | RU2 | .815 | .028 |
| **RS3** | **.101** | **.077** | **RI3** | **.216** | **.077** | **RF3** | **.359** | **.069** | **RU3** | **.328** | **.067** |
| RS4 | .875 | .025 | RI4 | .812 | .031 | RF4 | .88 | .021 | RU4 | .771 | .042 |
| CS1 | .893 | .019 | CI1 | .872 | .022 | CF1 | .915 | .015 | CU1 | .896 | .02 |
| CS2 | .906 | .016 | CI2 | .891 | .018 | CF2 | .896 | .019 | CU2 | .889 | .024 |
| CS3 | .94 | .01 | CI3 | .872 | .022 | CF3 | .934 | .013 | CU3 | .895 | .019 |
| CS4 | .932 | .014 | CI4 | .895 | .019 | CF4 | .907 | .023 | CU4 | .917 | .017 |
| COS1 | .771 | .034 | COI1 | .662 | .044 | COF1 | .741 | .039 | COU1 | .622 | .046 |
| COS2 | .866 | .025 | COI2 | .829 | .031 | COF2 | .886 | .024 | COU2 | .821 | .031 |
| COS3 | .861 | .029 | COI3 | .872 | .028 | COF3 | .902 | .025 | COU3 | .882 | .032 |
| **COS4** | **.117** | **.067** | **COI4** | **.276** | **.069** | **COF4** | **.192** | **.064** | **COU4** | **.154** | **.064** |
| TS1 | .797 | .038 | TI1 | .784 | .046 | TF1 | .753 | .05 | TU1 | .553 | .134 |
| TS2 | .712 | .054 | TI2 | .824 | .046 | TF2 | .777 | .054 | TU2 | .598 | .155 |
| TS3 | .779 | .047 | TI3 | .648 | .065 | TF3 | .78 | .05 | TU3 | .746 | .127 |
| **TS4** | **.284** | **.087** | **TI4** | **.207** | **.086** | **TF4** | **.439** | **.086** | **TU4** | **.413** | **.172** |
| OS1 | .792 | .038 | **OI1** | **.55** | **.059** | OF1 | .704 | .048 | **OU1** | **.565** | **.053** |
| OS2 | .87 | .026 | OI2 | .794 | .05 | OF2 | .802 | .039 | OU2 | .788 | .033 |
| OS3 | .901 | .018 | OI3 | .9 | .023 | OF3 | .851 | .036 | OU3 | .801 | .036 |
| OS4 | .858 | .023 | OI4 | .829 | .033 | OF4 | .871 | .024 | OU4 | .748 | .042 |
| RES1 | .795 | .036 | REI1 | .85 | .05 | REF1 | .802 | .041 | REU1 | .81 | .049 |
| RES2 | .874 | .03 | REI2 | .72 | .053 | REF2 | .763 | .047 | REU2 | .756 | .048 |
| RES3 | .785 | .041 | REI3 | .669 | .056 | REF3 | .798 | .045 | REU3 | .668 | .052 |
| **RES4** | **.05** | **.081** | **REI4** | **.14** | **.085** | **REF4** | **.135** | **.074** | **REU4** | **.075** | **.076** |

Note: Variables are named according to their DOI characteristic, context and their question number. R = Relative advantage, C = Compatibility, CO = Complexity, T = Trialability, O = Observability, RE = Reinvention. S = Social media filters, I = Image sensor, F = Financial systems, U = Phone unlock. Bolded items indicate poor fit.

**Table 3. Fit indices for sample 1 CFAs with updated items.**

| Model | χ² (df), p-value | RMSEA | CFI | TLI | SRMR |
|---|---|---|---|---|---|
| Phone Unlock | 223.244 (df = 137), p < .001 | 0.045 | 0.96 | 0.96 | 0.054 |
| Finances | 292.000 (df = 155), p < .001 | 0.053 | 0.96 | 0.95 | 0.049 |
| Social Media Filters | 219.114 (df = 155), p < .001 | 0.036 | 0.98 | 0.98 | 0.049 |
| Image Sensors | 246.513 (df = 137), p < .001 | 0.051 | 0.96 | 0.95 | 0.050 |

## Testing a nomological network

A nomological network is an interconnected system of theoretical constructs, observed variables and their relationships that allows researchers to test whether a measurement scale behaves as theory predicts [34]. In our context, testing the nomological network means examining whether our six DOI characteristics relate to other constructs (innovation use, algorithm awareness) in theoretically expected ways. This approach is crucial for scale validation because it demonstrates that our measures capture the intended theoretical constructs rather than other confounding factors. If our scale validly measures DOI characteristics, we should observe the pattern of relationships that DOI theory predicts. According

**Table 4. Factor loadings for the second CFA of study 1.**

| Social Media Filters | | | Image Sensors | | | | Financial Systems | | | Phone Unlock | | |
|---|---|---|---|---|---|---|---|---|---|---|---|---|
| param | est | se | param | est | se | | param | est | se | param | est | Se |
| RS1 | .893 | .019 | RI1 | .846 | .025 | | RF1 | .791 | .035 | RU1 | .782 | .032 |
| RS2 | .88 | .019 | RI2 | .813 | .034 | | RF2 | .75 | .031 | RU2 | .814 | .028 |
| RS4 | .875 | .025 | RI4 | .813 | .031 | | RF4 | .882 | .021 | RU4 | .774 | .041 |
| CS1 | .893 | .019 | CI1 | .871 | .022 | | CF1 | .915 | .015 | CU1 | .896 | .02 |
| CS2 | .906 | .016 | CI2 | .891 | .018 | | CF2 | .895 | .019 | CU2 | .889 | .024 |
| CS3 | .94 | .01 | CI3 | .872 | .022 | | CF3 | .934 | .013 | CU3 | .896 | .019 |
| CS4 | .932 | .014 | CI4 | .895 | .019 | | CF4 | .907 | .023 | CU4 | .917 | .017 |
| COS1 | .773 | .034 | COI1 | .662 | .044 | | COF1 | .741 | .039 | COU1 | .63 | .045 |
| COS2 | .866 | .026 | COI2 | .828 | .032 | | COF2 | .883 | .024 | COU2 | .821 | .031 |
| COS3 | .86 | .029 | COI3 | .874 | .028 | | COF3 | .905 | .024 | COU3 | .879 | .032 |
| TS1 | .799 | .037 | TI1 | .795 | .039 | | TF1 | .771 | .042 | TU1 | .657 | .065 |
| TS2 | .73 | .048 | TI2 | .835 | .041 | | TF2 | .82 | .042 | TU2 | .725 | .072 |
| TS3 | .754 | .046 | TI3 | .623 | .058 | | TF3 | .725 | .046 | TU3 | .597 | .075 |
| OS1 | .792 | .038 | OI2 | .792 | .05 | | OF1 | .704 | .048 | OU2 | .795 | .034 |
| OS2 | .869 | .026 | OI3 | .925 | .022 | | OF2 | .801 | .039 | OU3 | .816 | .036 |
| OS3 | .901 | .018 | OI4 | .81 | .035 | | OF3 | .851 | .036 | OU4 | .73 | .046 |
| OS4 | .858 | .023 | -- | -- | -- | | OF4 | .871 | .024 | -- | -- | -- |
| RES1 | .796 | .036 | REI1 | .858 | .05 | | REF1 | .804 | .042 | REU1 | .798 | .049 |
| RES2 | .873 | .03 | REI2 | .712 | .053 | | REF2 | .76 | .047 | REU2 | .767 | .046 |
| RES3 | .784 | .041 | REI3 | .665 | .057 | | REF3 | .798 | .045 | REU3 | .67 | .052 |

Note: Variables are named according to their DOI characteristic, context, and question number. R = Relative advantage, C = Compatibility, CO = Complexity, T = Trialability, O = Observability, RE = Reinvention. S = Social media filters, I = Image sensor, F = Financial systems, U = Phone unlock.

to DOI theory, relative advantage, compatibility, complexity (reversed), trialability, observability, and reinvention should be positively associated with innovation use (in this case, the use of image recognition technology; H1) [2]. If these six DOI characteristics are not positively associated with innovation use, it could suggest measurement issues, contextual factors unique to image recognition technology, or potential moderating variables that influence these relationships.

It is also important to ensure that the scale does not measure similar, but potentially confounding constructs [35], which we call external discriminant validity. One such construct that is different from the six innovation characteristics in our scale is the user's understanding of how the technology works, which, in the case of software-based innovations, is the user's *algorithm awareness*. Algorithm awareness refers to a user's awareness of the underlying mechanisms and factors that influence how a particular software functions. While this awareness might inform users' perceptions of an innovation, it is distinct from the six DOI characteristics as they have to do with users' subjective interpretation of the innovation. Therefore, we propose that relative advantage, compatibility, complexity, trialability, observability, and reinvention will not be more than moderately correlated (i.e., $r > 0.30$; [37] with algorithmic awareness (H2). We chose this threshold based on Cohen's (1988) guidelines for interpreting effect sizes in social sciences, where correlations of 0.10, 0.30, and 0.50 are considered small, medium, and large, respectively.

Finally, to assess construct validity, we will examine perceptions of relative advantage, compatibility, complexity, trialability, observability, and reinvention across various software innovations. As mentioned previously, we expect the constructs associated with DOI to vary across different contexts, requiring customization. Therefore, we expect individuals to perceive these attributes differently depending on the innovation (H3). If we observe no significant differences between contexts, it may suggest that our scale lacks sensitivity to the unique characteristics of different software innovations or that the software innovations are perceived as having similar attributes.

## Methods

### Participants

We recruited 851 participants from Prolific, a web-based survey platform, aiming for a sample representative of the U.S. population in June of 2023. The sample was diverse in terms of gender (41% women, 56% men, 3% agender, transgender, non-binary, or unspecified), age (M = 36.17, SD = 12.47), and ethnicity (22% Asian, 26.6% Black/African American, 18% Latino, 24% White, 6% mixed White/Latino, 3.4% mixed race). Participants reported a median income between $50,000-$59,999, and the median education level was a bachelor's degree. For more information about Prolific participants, see [36,37].

This research was reviewed and approved by the University of California, Santa Barbara Human Subjects Committee (Protocol #4-23-0221). The study was determined to be exempt under Category 2 of 45 CFR 46.104(d), which covers survey procedures involving adult participants. All participants provided written informed consent prior to participation.

### Procedure

We asked participants about their perceptions of the six DOI characteristics across the same four image recognition-based software innovations in Study 1: phone unlock, social media filters, finances, and image sensors. We conducted CFA with maximum likelihood estimation using a variance-covariance matrix on Mplus [30]. We specified the CFA model with the same structure used in Study 1. We used standard fit criteria, considering models with a SRMR ≤ .08, a CFI and TLI ≥ .95, and an RMSEA < 0.08 a good fit [31–33].

### Measures

All measures of the perceptual DOI characteristics in Study 2 are the same as Study 1 with some exceptions, which we noted in the results section of Study 1. In addition to the changes previously described, we dropped parentheticals from all questions to enhance question clarity, for example the question "[innovation] fits (would fit) well with the way that I like to [task]" has been changed to "[innovation] fits well with the way that I like to [task]". Study 2 included two additional measures: innovation use and algorithm awareness. Table 5 provides Cronbach's alpha, means, and standard deviations for the DOI measures.

### Innovation use

We assessed the use of image recognition technology by asking participants how often they use 10 facial and image recognition technologies related to the four innovations (e.g., "In general, how often do you use the following technologies: e.g., automatic water dispenser"). We measured responses on a 5-point scale from *never to always*. (Phone unlock M = 2.93, SD = 1.69, finances M = 2.54, SD = 1.62, social media filters, M = 2.44, SD = 1.30, image sensors, M = 3.01, SD = 0.81).

### Algorithm awareness

We assessed participants' algorithm awareness by asking how 10 different factors—5 factors for image sensing technology and 5 for facial recognition technology—influence the output of facial recognition and image sensing technology. Algorithm awareness was divided into two categories: facial recognition algorithm awareness and image sensing algorithm awareness. Both facial recognition technology and image sensing technology are influenced by similar, but not always the same, underlying factors. For example, facial recognition technology is influenced by physical features such as face shape, while image sensing technology is not. We adapted the scale prompt, "Generally speaking, how much INFLUENCE do you think the following factors have on the output or results of a [facial recognition or image sensing] algorithm," from [38], basing the answers on current literature that discusses the factors influencing facial recognition technology and

**Table 5. DOI Measure Descriptive Statistics Study 2.**

| | Relative Advantage | | | Compatibility | | | Complexity | | |
|---|---|---|---|---|---|---|---|---|---|
| | *α* | *M* | *SD* | *α* | *M* | *SD* | *α* | *M* | *SD* |
| Phone Unlock | .93 | 3.29 | 1.36 | .97 | 3.39 | 1.43 | .89 | 3.91 | 1.00 |
| Finances | .93 | 3.05 | 1.35 | .97 | 3.08 | 1.44 | .92 | 3.63 | 1.12 |
| Social Media | .90 | 2.72 | 1.23 | .96 | 2.68 | 1.36 | .89 | 3.75 | 1.00 |
| Image Sensors | .90 | 3.55 | 1.08 | .94 | 3.84 | 1.06 | .87 | 3.96 | 0.93 |
| | **Observability** | | | **Trialability** | | | **Reinvention** | | |
| | *α* | *M* | *SD* | *α* | *M* | *SD* | *α* | *M* | *SD* |
| Phone Unlock | .92 | 3.64 | 1.09 | .84 | 3.79 | 1.13 | .80 | 3.74 | 1.07 |
| Finances | .92 | 2.96 | 1.20 | .88 | 3.27 | 1.28 | .82 | 3.09 | 1.20 |
| Social Media | .90 | 3.73 | 1.04 | .83 | 3.72 | 1.07 | .79 | 3.62 | 1.02 |
| Image Sensors | .91 | 4.09 | 0.94 | .80 | 3.87 | 0.95 | .77 | 3.88 | 0.96 |

image sensing technology. Example items include: "Lighting conditions of the environment" and "Other phenotypical features, such as your face shape." We scored responses on a 5-point scale from (1) *strongly disagree* to (5) *strongly agree.* (Facial Recognition Algorithm Awareness, α = .68, *M* = 3.53*, SD* = 0.78; Image Sensing Algorithm Awareness, α = 0.68, *M* = 3.62*, SD* = 0.78).

## Results

### CFA

As seen in Table 6, the CFAs on the second sample had a good fit, with an SRMR ≤ .08, a CFI and TLI ≥ . 95, and an RMSEA < 0.08 for all four contexts (refer to Table 7 for factor loadings).

### Internal discriminant validity

To test for internal discriminant validity, we examined the correlations between differing factors**,** which must not be too high [39]. Generally, a correlation of.85 or larger in absolute value indicates poor discriminant validity [33].

Overall, the scale had good discriminant validity, with all factor correlations within each context being less than.85 (see Table 8), with the notable exception of relative advantage and compatibility in the finances and phone unlock contexts, which are correlated at a level of.928 and.904 respectively. In cases where two factors correlate more than.85, it is acceptable to consider them as part of a single scale, even though they function effectively as separate constructs. Given the high correlations between relative advantage and compatibility in the finances and phone unlock contexts, we conducted an additional CFA for each of these two contexts to see if combining relative advantage and compatibility would significantly improve model fit. The resulting fit statistics were similar to, and in the case of phone unlock, slightly worse than having relative advantage as separate factors. (Phone Unlock: $\chi^2(67)$ = 693.681 (p < .001), RMSEA = .068, CFI = .946, TLI = .935 SRMR = .047; Finances: $\chi^2(67)$ = 594.921 (p < .001), RMSEA = .062, CFI = .96, TLI = .95, SRMR = .041).

**Table 6. Fit indices for sample 2 CFAs.**

| Technology context | $\chi^2$ (df), p-value | RMSEA | CFI | TLI | SRMR |
|---|---|---|---|---|---|
| Phone Unlock | 525.751 (137), p < .001 | .058 | .967 | .959 | .046 |
| Finances | 338.462 (137), p < .001 | .042 | .983 | .979 | .032 |
| Social Media Filters | 433.494 (137), p < .001 | .050 | .972 | .964 | .05 |
| Image Sensors | 477.859 (137), p < .001 | .054 | .961 | .951 | .048 |

**Table 7. Factor Loadings for second study CFAs.**

| param | est | se | param | est | se | param | est | se | param | est | se |
|---|---|---|---|---|---|---|---|---|---|---|---|
| RS1 | .855 | .014 | RI1 | .833 | .016 | RF1 | .884 | .011 | RU1 | .855 | .014 |
| RS2 | .835 | .015 | RI2 | .893 | .011 | RF2 | .887 | .011 | RU2 | .923 | .009 |
| RS3 | -- | -- | RI3 | -- | -- | RF3 | -- | -- | RU3 | -- | -- |
| RS4 | .934 | .008 | RI4 | .904 | .011 | RF4 | .953 | .006 | RU4 | .943 | .008 |
| CS1 | .94 | .008 | CI1 | .91 | .011 | CF1 | .945 | .006 | CU1 | .945 | .006 |
| CS2 | .933 | .007 | CI2 | .906 | .01 | CF2 | .938 | .007 | CU2 | .933 | .008 |
| CS3 | .955 | .005 | CI3 | .917 | .009 | CF3 | .961 | .005 | CU3 | .953 | .005 |
| CS4 | .953 | .007 | CI4 | .942 | .008 | CF4 | .965 | .004 | CU4 | .968 | .004 |
| COS1 | .78 | .021 | COI1 | .758 | .021 | COF1 | .849 | .016 | COU1 | .751 | .021 |
| COS2 | .877 | .018 | COI2 | .887 | .014 | COF2 | .906 | .011 | COU2 | .902 | .011 |
| COS3 | .908 | .012 | COI3 | .891 | .014 | COF3 | .931 | .01 | COU3 | .926 | .011 |
| COS4 | -- | -- | COI4 | -- | -- | COF4 | -- | -- | COU4 | -- | -- |
| TS1 | .787 | .024 | TI1 | .738 | .027 | TF1 | .795 | .02 | TU1 | .753 | .025 |
| TS2 | .792 | .022 | TI2 | .861 | .023 | TF2 | .888 | .013 | TU2 | .864 | .018 |
| TS3 | .815 | .022 | TI3 | .679 | .033 | TF3 | .839 | .016 | TU3 | .793 | .024 |
| TS4 | -- | -- | TI4 | -- | -- | TF4 | -- | -- | TU4 | -- | -- |
| OS1 | -- | -- | OI1 | -- | -- | OF1 | -- | -- | OU1 | -- | -- |
| OS2 | .858 | .016 | OI2 | .871 | .018 | OF2 | .87 | .013 | OU2 | .883 | .014 |
| OS3 | .911 | .013 | OI3 | .907 | .015 | OF3 | .915 | .012 | OU3 | .916 | .011 |
| OS4 | .854 | .018 | OI4 | .874 | .018 | OF4 | .908 | .012 | OU4 | .879 | .015 |
| RES1 | .838 | .017 | REI1 | .834 | .02 | REF1 | .867 | .017 | REU1 | .872 | .016 |
| RES2 | .884 | .016 | REI2 | .858 | .019 | REF2 | .893 | .013 | REU2 | .91 | .014 |
| RES3 | .843 | .017 | REI3 | .796 | .023 | REF3 | .873 | .015 | REU3 | .825 | .021 |
| RES4 | -- | -- | REI4 | -- | -- | REF4 | -- | -- | REU4 | -- | -- |

Note: Variables are named according to their DOI characteristic, context, and question number. R = Relative advantage, C = Compatibility, CO = Complexity, T = Trialability, O = Observability, RE = Reinvention. S = Social media filters, I = Image sensor, F = Financial systems, U = Phone unlock.

Moreover, conceptually and theoretically, these factors are distinct; the scale performs well with them being treated as different entities and they only have a correlation > .85 in two out of the four technological contexts. The high correlation suggests a strong relationship but does not diminish their individual theoretical significance or practical utility in measuring separate aspects of the construct. Thus, we have decided to leave them as separate factors going forward.

## Relationships within a nomological network

We assessed predictive validity (H1) by correlating scores from the six DOI characteristics with use of image recognition technology. For each model, we used either relative advantage, compatibility, complexity, observability, trialability, or reinvention (independent variables) to predict the use of facial recognition software to unlock a phone, facial recognition software to unlock a financial account, social media facial filters, or image sensors (dependent variables) using a linear regression. Consistent with our prediction, relative advantage, compatibility, trialability, observability, complexity, and reinvention significantly positively predict the use of image recognition-based software innovations (See Table 9).

To assess external discriminant validity (H2), we examined correlations between the six DOI characteristics and algorithm awareness. We ran a correlation analysis to examine the relationships between the facial recognition contexts (i.e., phone unlock, social media filters, and finances) and facial recognition algorithm awareness. Similarly, we analyzed the image sensing context and its correlation with image sensing algorithm awareness.

**Table 8. Correlations between factors sample 2.**

**Image Sensors**

| | 1 | 2 | 3 | 4 | 5 | 6 |
|---|---|---|---|---|---|---|
| Relative Advantage 1 | -- | | | | | |
| Compatibility 2 | .795 | -- | | | | |
| Complexity 3 | .616 | .704 | -- | | | |
| Trialability 4 | .488 | .547 | .632 | -- | | |
| Observability 5 | .374 | .514 | .594 | .576 | -- | |
| Reinvention 6 | .177 | .086 | −.060 | .174 | .015 | -- |

**Social Media Filters**

| | 1 | 2 | 3 | 4 | 5 | 6 |
|---|---|---|---|---|---|---|
| Relative Advantage 1 | -- | | | | | |
| Compatibility 2 | .834 | -- | | | | |
| Complexity 3 | .421 | .439 | -- | | | |
| Trialability 4 | .398 | .418 | .720 | -- | | |
| Observability 5 | .315 | .286 | .626 | .578 | -- | |
| Reinvention 6 | .638 | .602 | .190 | .291 | .231 | -- |

**Finances**

| | 1 | 2 | 3 | 4 | 5 | 6 |
|---|---|---|---|---|---|---|
| Relative Advantage 1 | -- | | | | | |
| Compatibility 2 | **.928** | -- | | | | |
| Complexity 3 | .698 | .720 | -- | | | |
| Trialability 4 | .738 | .760 | .776 | -- | | |
| Observability 5 | .499 | .495 | .489 | .534 | -- | |
| Reinvention 6 | .373 | .313 | .170 | .346 | .458 | -- |

**Phone Unlock**

| | 1 | 2 | 3 | 4 | 5 | 6 |
|---|---|---|---|---|---|---|
| Relative Advantage 1 | -- | | | | | |
| Compatibility 2 | **.904** | -- | | | | |
| Complexity 3 | .594 | .658 | -- | | | |
| Trialability 4 | .658 | .616 | .730 | -- | | |
| Observability 5 | .434 | .465 | .579 | .551 | -- | |
| Reinvention 6 | .288 | .283 | .025 | .200 | .175 | -- |

None of the facial recognition factors were significantly associated with facial recognition algorithm awareness. Only trialability and observability of image sensing technologies were significantly correlated with image recognition algorithm awareness (See Table 10), with both having notably small effect sizes (i.e., $r < 0.30$; Cohen, 1988), thus providing support for the scale's external discriminant validity and supporting H2.

Finally, to test H3, we conducted ANOVAs with Tukey HSD post hoc tests to determine if there were significant mean differences between the various software innovations. Each ANOVA tested the mean differences between one DOI attribute and perceptions of four different software innovations: image sensors, finance applications, phone unlocking features, and social media filters.

We observed a significant effect of innovation type on relative advantage ($F_{(3, 3379)} = 66.58$, $p < .000$, $\eta2 = .056$) with significant differences in perceptions of relative advantage between all innovations: image sensors and finances ($p < .001$), phone unlock and finances ($p < .001$), social media and finances ($p < .001$), phone unlock and image sensors ($p < .001$), social media and image sensors ($p < .001$), and social media and phone unlock ($p < .001$).

**Table 9. Study 2: Linear Regression Predicting Use of Four Software-Based Innovations by Diffusion of Innovations (DOI) Characteristics.**

| | Relative Advantage | | Compatibility | | Complexity | |
|---|---|---|---|---|---|---|
| | b | p | b | p | b | p |
| Phone Unlock | .960 | <.001 | .899 | <.001 | .819 | <.001 |
| Finances | .889 | <.001 | .834 | <.001 | .794 | <.001 |
| Social Media Filters | .665 | <.001 | .658 | <.001 | .449 | <.001 |
| Image Sensors | .201 | <.001 | .232 | <.001 | .158 | <.001 |
| | **Observability** | | **Trialability** | | **Reinvention** | |
| | b | p | b | p | b | p |
| Phone Unlock | .542 | <.001 | .711 | <.001 | .346 | <.001 |
| Finances | .512 | <.001 | .785 | <.001 | .354 | <.001 |
| Social Media Filters | .274 | <.001 | .363 | <.001 | .582 | <.001 |
| Image Sensors | .158 | <.001 | .131 | <.001 | .140 | <.001 |

*Note. All betas are unstandardized.*

**Table 10. Correlations between DOI Characteristics and Algorithm Awareness.**

| DOI Characteristic | Facial recognition algorithm awareness | Image recognition algorithm awareness |
|---|---|---|
| Relative Advantage Phone Unlock | −.02 | |
| Relative Advantage Social Media Filters | .07 | |
| Relative Advantage Finances | −.01 | |
| Relative Advantage Image Sensors | | .06 |
| Compatibility Phone Unlock | −.05 | |
| Compatibility Social Media Filters | .08 | |
| Compatibility Finances | −.04 | |
| Compatibility Image Sensors | | .11 |
| Complexity Phone Unlock | .00 | |
| Complexity Social Media Filters | .08 | |
| Complexity Finances | −.01 | |
| Complexity Image Sensors | | .08 |
| Trialability Phone Unlock | .07 | |
| Trialability Social Media Filters | .06 | |
| Trialability Finances | .02 | |
| Trialability Image Sensors | | **.13*** |
| Observability Phone Unlock | .10 | |
| Observability Social Media Filters | .11 | |
| Observability Finances | .09 | |
| Observability Image Sensors | | **.18*** |
| Reinvention Phone Unlock | .11 | |
| Reinvention Social Media Filters | .09 | |
| Reinvention Finances | .10 | |
| Reinvention Image Sensors | | .07 |

*\* = p <.05.*

We observed a significant effect of innovation type on compatibility (F(3,3366) = 114.7, p<.000, η2 = .093) with significant differences in perceptions of compatibility between all innovations: image sensors and finances (*p<.001*), phone unlock and finances (*p<.001*), social media and finances (*p<.001*), phone unlock and image sensors (*p<.001*), social media and image sensors (*p<.001*), and social media and phone unlock (*p<.001*).

We observed a significant effect of innovation type on complexity (F(3, 3377) = 18.73, *p<.000*, $\eta^2$ = .016) with significant differences between perceptions of complexity between image sensors and finances (*p<.001*), phone unlock and finances (*p<.001*), social media and image sensors (*p<.001*), and social media and phone unlock (*p<.01*). There were no significant differences between social media and finances (*p* = .08) or phone unlock and image sensors (*p* = .73).

We observed a significant effect of innovation type on trialability (F(3, 3382) = 49.08, *p<.000*, $\eta^2$ = .042) with significant mean differences between image sensors and finances (*p<.001*), phone unlock and finances (*p<.001*), social media and finances (*p<.001*), and social media and image sensors (*p<.05*). There were no significant differences between phone unlock and image sensors and finances (*p* = .46) or social media and phone unlock (*p* = .60).

We observed a significant effect of innovation type on observability (F(3, 3385) = 166.1, *p<.000*, $\eta^2$ = 0.128) with significant mean differences between all pairs of variables: image sensors and finances (*p<.001*), phone unlock and finances (*p<.001*), social media and finances (*p<.001*), phone unlock and image sensors (*p<.001*), social media and image sensors (*p<.001*), except for social media and phone unlock (*p* = .28).

We observed a significant effect of innovation type on reinvention (F(3, 3379) = 3.423, *p* = .016, $\eta^2$ = .003) with significant mean differences between image sensors and finances (*p<.05*), but not any of the other variable pairs: phone unlock and finances (*p* = .99), social media and finances (*p* = .25), phone unlock and image sensors (*p* = .058), social media and image sensors (*p* = .82), social media and phone unlock (*p* = .34).

## Study 2 discussion

Study 2 supports the six-factor measure of the perceptual DOI characteristics. The fit statistics of the CFAs meet the standard fit criteria and generally have good discriminant validity. This stance is further corroborated by all item loadings being greater than.6, and Cronbach's alphas exceeding.75 for all six perceptual characteristics. Furthermore, study 2 offers support for the validity of the scale within a nomological network. All factors significantly predicted the use of software-based innovations (H1) and were not significantly correlated with algorithm awareness (H2). Participants also displayed different mean ratings for relative advantage, compatibility, complexity, trialability, observability, and reinvention across various innovations (H3).

## Study 3

Study 3 replicates the analysis from Study 2. Due to significant differences between the populations in our first and second samples—both in sample size and demographic distribution—and the absence of discriminant validity assessments in sample 1 or an evaluation of the nomological network of variables related to the diffusion of innovations, we collected a third sample to replicate the results of Study 2.

## Methods

### Participants

We recruited 843 participants for this study in May of 2024. The sample was diverse in terms of gender (48% women, 51% men, 1% non-binary or unspecified), age (M = 36.75, SD = 11.4), and ethnicity (22.1% Asian, 27.8% Black/African American, 18% Latino, 23.2% White, 5.2% mixed White/Latino, 3.7% mixed race). Participants reported a median income between $60,000-$69,999, and the median education level was a bachelor's degree.

This research was reviewed and approved by the University of California, Santa Barbara Human Subjects Committee (Protocol #4-23-0221). The study was determined to be exempt under Category 2 of 45 CFR 46.104(d), which covers survey procedures involving adult participants. All participants provided written informed consent prior to participation.

### Procedure

Study 3 follows the same procedure as study 2.

### Measures

All measures in Study 3 are the same as Study 2. Table 11 provides Cronbach's alpha, means, and standard deviations for all measures in study 3.

## Results

### CFA

As seen in Table 13, the results of the CFA analysis suggest satisfactory models on all goodness of fit statistics with the SRMR less than.08, the RMSEA falling below.08, and TLI/CFI values falling above or equal to.95 across all models except image sensors, where the CFI and TLI are.945 and.931 respectively. Although these values are slightly below the ideal threshold, values above 0.90 are still considered indicative of an acceptable model fit (Brown, 2006), especially when supported by other acceptable fit indices. Fit indices should be interpreted as continuous indicators of model fit rather than as rigid thresholds for accepting or rejecting a model. Refer to Table 12 for updated factor loadings.

### Internal discriminant validity

The scale has good internal discriminant validity, with all factor correlations within each context being less than.85 (see Table 14), with the exception of relative advantage and compatibility in the finances and phone unlock contexts, as with sample 2, which are correlated at a level of.945 and.918 respectively.

**Table 11. DOI Measure Descriptive Statistics Study 3.**

| Relative Advantage | | | | Compatibility | | | | Complexity | | | |
|---|---|---|---|---|---|---|---|---|---|---|---|
| | *α* | *M* | *SD* | | *α* | *M* | *SD* | | *α* | *M* | *SD* |
| Phone Unlock | .92 | 3.57 | 1.28 | | .97 | 3.66 | 1.34 | | .88 | 4.02 | 0.93 |
| Finances | .92 | 3.29 | 1.30 | | .98 | 3.31 | 1.40 | | .91 | 3.75 | 1.05 |
| Social Media | .88 | 2.77 | 1.20 | | .96 | 2.80 | 1.32 | | .89 | 3.75 | 0.99 |
| Image Sensors | .91 | 3.54 | 1.09 | | .95 | 3.86 | 1.01 | | .87 | 3.96 | 0.92 |
| **Observability** | | | | **Trialability** | | | | **Reinvention** | | | |
| | *α* | *M* | *SD* | | *α* | *M* | *SD* | | *α* | *M* | *SD* |
| Phone Unlock | .90 | 3.72 | 1.04 | | .80 | 3.88 | 1.03 | | .90 | 2.42 | 1.21 |
| Finances | .91 | 3.09 | 1.16 | | .84 | 3.41 | 1.18 | | .89 | 2.39 | 1.17 |
| Social Media | .89 | 3.81 | 0.96 | | .82 | 3.74 | 1.03 | | .88 | 2.57 | 1.16 |
| Image Sensors | .90 | 4.14 | 0.89 | | .73 | 3.80 | 0.92 | | .85 | 2.67 | 1.15 |
| **Innovation Use** | | | | **Algorithm Awareness** | | | | | | | |
| | | *M* | *SD* | | | | | | *α* | *M* | *SD* |
| Phone Unlock | | 3.37 | 1.63 | Facial Recognition Image Sensing | | | | | .67 | 3.48 | 0.76 |
| Finances | | 2.97 | 1.68 | | | | | | .60 | 3.64 | 0.69 |
| Social Media | | 2.53 | 1.30 | | | | | | | | |
| Image Sensors | | 3.20 | 0.78 | | | | | | | | |

**Table 12. Factor Loadings for Study 3 CFAs.**

| Social Media Filters | | | Image Sensors | | | Finances | | | Phone Unlock | | |
|---|---|---|---|---|---|---|---|---|---|---|---|
| param | est | se | param | est | se | param | est | Se | param | est | se |
| RS1 | .833 | .016 | RI1 | .842 | .017 | RF1 | .895 | .01 | RU1 | .857 | .015 |
| RS2 | .834 | .015 | RI2 | .897 | .011 | RF2 | .883 | .011 | RU2 | .915 | .01 |
| RS3 | -- | -- | RI3 | -- | -- | RF3 | -- | -- | RU3 | -- | -- |
| RS4 | .901 | .012 | RI4 | .907 | .013 | RF4 | .916 | .01 | RU4 | .924 | .009 |
| CS1 | .916 | .01 | CI1 | .918 | .008 | CF1 | .949 | .006 | CU1 | .945 | .007 |
| CS2 | .924 | .008 | CI2 | .898 | .011 | CF2 | .945 | .006 | CU2 | .928 | .01 |
| CS3 | .936 | .007 | CI3 | .914 | .009 | CF3 | .957 | .005 | CU3 | .956 | .005 |
| CS4 | .954 | .005 | CI4 | .93 | .008 | CF4 | .967 | .004 | CU4 | .95 | .005 |
| COS1 | .798 | .02 | COI1 | .77 | .019 | COF1 | .831 | .017 | COU1 | .766 | .021 |
| COS2 | .884 | .013 | COI2 | .856 | .018 | COF2 | .897 | .012 | COU2 | .886 | .015 |
| COS3 | .907 | .012 | COI3 | .898 | .013 | COF3 | .933 | .009 | COU3 | .893 | .015 |
| COS4 | -- | -- | COI4 | -- | -- | COF4 | -- | -- | COU4 | -- | -- |
| TS1 | .804 | .022 | TI1 | .648 | .041 | TF1 | .712 | .026 | TU1 | .707 | .032 |
| TS2 | .734 | .024 | TI2 | .766 | .04 | TF2 | .882 | .013 | TU2 | .828 | .024 |
| TS3 | .808 | .022 | TI3 | .672 | .054 | TF3 | .796 | .021 | TU3 | .717 | .033 |
| TS4 | -- | -- | TI4 | -- | -- | TF4 | -- | -- | TU4 | -- | -- |
| OS1 | -- | -- | OI1 | -- | -- | OF1 | -- | -- | OU1 | -- | -- |
| OS2 | .853 | .018 | OI2 | .899 | .014 | OF2 | .848 | .015 | OU2 | .84 | .018 |
| OS3 | .891 | .016 | OI3 | .903 | .015 | OF3 | .914 | .01 | OU3 | .906 | .012 |
| OS4 | .827 | .021 | OI4 | .818 | .024 | OF4 | .892 | .013 | OU4 | .886 | .015 |
| RES1 | .824 | .018 | REI1 | .859 | .02 | REF1 | .862 | .017 | REU1 | .867 | .017 |
| RES2 | .863 | .016 | REI2 | .8 | .022 | REF2 | .855 | .018 | REU2 | .893 | .015 |
| RES3 | .844 | .016 | REI3 | .77 | .024 | REF3 | .867 | .014 | REU3 | .854 | .017 |
| RES4 | -- | -- | REI4 | -- | -- | REF4 | -- | -- | REU4 | -- | -- |

Note: Variables are named according to their DOI characteristic, context, and question number. R = Relative advantage, C = Compatibility, CO = Complexity, T = Trialability, O = Observability, RE = Reinvention. S = Social media filters, I = Image sensor, F = Financial systems, U = Phone unlock.

**Table 13. Fit indices for sample 3 CFAs.**

| Technology context | χ² (df), p-value | RMSEA | CFI | TLI | SRMR |
|---|---|---|---|---|---|
| Phone Unlock | 466.929 (137), p < .001 | .053 | .968 | .96 | .045 |
| Finances | 441.297 (137), p < .001 | .051 | .975 | .968 | .039 |
| Social Media Filters | 502.310 (137), p < .001 | .056 | .964 | .956 | .052 |
| Image Sensors | 649.236 (137), p < .001 | .067 | **.945** | **.931** | .054 |

## Relationships within a nomological network

We ran the same analysis in sample 3 as in sample 2 to assess predictive validity. Consistent with our prediction, as seen in Table 15, relative advantage, compatibility, trialability, observability, complexity, and reinvention significantly positively predicted the use of image recognition-based software innovations.

We ran the same analysis in sample 3 as in sample 2 to assess external discriminant validity. As seen in Table 16, reinvention was significantly positively associated with facial recognition algorithm awareness for all innovations, and observability of image sensors was significantly correlated with image recognition algorithm awareness. All other variables were

**Table 14. Study 3: Correlations Between Factors.**

**Image Sensors**

| | 1 | 2 | 3 | 4 | 5 | 6 |
|---|---|---|---|---|---|---|
| Relative Advantage 1 | -- | | | | | |
| Compatibility 2 | .806 | -- | | | | |
| Complexity 3 | .672 | .721 | -- | | | |
| Trialability 4 | .520 | .608 | .608 | -- | | |
| Observability 5 | .331 | .491 | .553 | .627 | -- | |
| Reinvention 6 | .177 | .093 | −.055 | .228 | .061 | -- |

**Social Media Filters**

| | 1 | 2 | 3 | 4 | 5 | 6 |
|---|---|---|---|---|---|---|
| Relative Advantage 1 | -- | | | | | |
| Compatibility 2 | .849 | -- | | | | |
| Complexity 3 | .452 | .464 | -- | | | |
| Trialability 4 | .453 | .445 | .811 | -- | | |
| Observability 5 | .311 | .305 | .569 | .569 | -- | |
| Reinvention 6 | .665 | .591 | .294 | .338 | .217 | -- |

**Finances**

| | 1 | 2 | 3 | 4 | 5 | 6 |
|---|---|---|---|---|---|---|
| Relative Advantage 1 | -- | | | | | |
| Compatibility 2 | **.945** | -- | | | | |
| Complexity 3 | .735 | .734 | -- | | | |
| Trialability 4 | .752 | .773 | .724 | -- | | |
| Observability 5 | .493 | .468 | .429 | .552 | -- | |
| Reinvention 6 | .343 | .293 | .124 | .326 | .431 | -- |

**Phone Unlock**

| | 1 | 2 | 3 | 4 | 5 | 6 |
|---|---|---|---|---|---|---|
| Relative Advantage 1 | -- | | | | | |
| Compatibility 2 | **.918** | -- | | | | |
| Complexity 3 | .692 | .695 | -- | | | |
| Trialability 4 | .602 | .614 | .68 | -- | | |
| Observability 5 | .356 | .348 | .448 | .498 | -- | |
| Reinvention 6 | .231 | .180 | .036 | .211 | .237 | -- |

not significantly correlated to either facial recognition algorithm awareness or image sensor algorithm awareness. As with sample 2, all significant correlations had small effect sizes having (i.e., r<0.30; [37]), thus providing support for H3.

We applied the same analysis to test H3 as we did in Study 2.

We observed a significant effect of innovation type on relative advantage (F(3, 3350) = 77.48, $p < .000$, $\eta^2 = .055$) with significant differences in perceptions of relative advantage between all innovations ([image sensors & finances] $p < .001$; [phone unlock & finances] $p < .001$; [social media & finances] $p < .001$; [social media & image sensors] $p < .001$; [social media & phone unlock] $p < .001$) except for phone unlock and image sensors ($p = .98$).

We observed a significant effect of innovation type on compatibility ($F(3,3341) = 110.7$, $p < .000$, $\eta^2 = .090$) with significant differences in perceptions of compatibility between all innovations: image sensors and finances ($p < .001$), phone unlock and finances ($p < .001$), social media and finances ($p < .001$), social media and image sensors ($p < .001$); social media and phone unlock ($p < .001$), and phone unlock and image sensors ($p < .01$).

**Table 15. Study 3: Results of Linear Regression Where Each DOI Characteristic Was Used to Predict the Use of Each of the Four Software-Based Innovations.**

| | Relative Advantage | | Compatibility | | Complexity | |
|---|---|---|---|---|---|---|
| | *b* | *p* | *b* | *p* | *b* | *p* |
| Phone Unlock | .990 | *<.001* | .956 | *<.001* | .938 | *<.001* |
| Finances | .960 | *<.001* | .929 | *<.001* | .886 | *<.001* |
| Social Media Filters | .700 | *<.001* | .667 | *<.001* | .459 | *<.001* |
| Image Sensors | .239 | *<.001* | .268 | *<.001* | .195 | *<.001* |
| | Observability | | Trialability | | Reinvention | |
| | *b* | *p* | *b* | *p* | *b* | *p* |
| Phone Unlock | .445 | *<.001* | .639 | *<.001* | .227 | *<.001* |
| Finances | .554 | *<.001* | .818 | *<.001* | .388 | *<.001* |
| Social Media Filters | .335 | *<.001* | .468 | *<.001* | .587 | *<.001* |
| Image Sensors | .201 | *<.001* | .187 | *<.001* | .162 | *<.001* |

*Note. All betas are unstandardized.*

**Table 16. Study 3: Correlations between DOI Characteristics and Algorithm Awareness.**

| DOI Characteristic | Facial recognition algorithm awareness | Image sensor algorithm awareness |
|---|---|---|
| Relative Advantage Phone Unlock | −.08 | |
| Relative Advantage Social Media Filters | .09 | |
| Relative Advantage Finances | −.07 | |
| Relative Advantage Image Sensors | | .05 |
| Compatibility Phone Unlock | −.09 | |
| Compatibility Social Media Filters | .06 | |
| Compatibility Finances | −.06 | |
| Compatibility Image Sensors | | .09 |
| Complexity Phone Unlock | −.08 | |
| Complexity Social Media Filters | .06 | |
| Complexity Finances | −.05 | |
| Complexity Image Sensors | | .02 |
| Trialability Phone Unlock | .00 | |
| Trialability Social Media Filters | .09 | |
| Trialability Finances | −.02 | |
| Trialability Image Sensors | | .10 |
| Observability Phone Unlock | .06 | |
| Observability Social Media Filters | .11 | |
| Observability Finances | .09 | |
| Observability Image Sensors | | **.17*** |
| Reinvention Phone Unlock | **.12*-** | |
| Reinvention Social Media Filters | **.14** | |
| Reinvention Finances | **.13** | |
| Reinvention Image Sensors | | .10 |

*\* = p <.05 \*\* = p <.01 \*\*\* = p <.01.*

We observed a significant effect of innovation type on complexity (F(3, 3352) = 17.04, $p < .000$, $\eta^2 = .015$) with significant differences between perceptions of image sensors and finances ($p < .001$), phone unlock and finances ($p < .001$), social media and image sensors ($p < .001$), and social media and phone unlock ($p < .001$). However, there were no significant differences between social media and finances ($p = .99$) or phone unlock and image sensors ($p = .59$).

We observed a significant effect of innovation type on trialability (F(3, 3352) = 32.24, $p < .000$, $\eta^2 = .028$) with significant mean differences between image sensors and finances ($p < .001$), phone unlock and finances ($p < .001$), social media and finances ($p < .001$), and social media and phone unlock ($p < .05$). However, there were no significant differences between phone unlock and image sensors ($p = .49$) or social media and image sensors ($p = .60$).

We observed a significant effect of innovation type on observability (F(3, 3356) = 155.8, $p < .000$, $\eta^2 = .12$) with significant mean differences between all pairs of variables: image sensors and finances ($p < .001$), phone unlock and finances ($p < .001$), social media and finances ($p < .001$), phone unlock and image sensors ($p < .001$), social media and image sensors ($p < .001$), except social media and phone unlock ($p = .26$).

We observed a significant effect of innovation type on reinvention (F(3, 3352) = 10.19, $p = <.0000$, $\eta^2 = .009$) with significant mean differences between image sensors and finances ($p < .001$), social media and finances ($p < .05$), phone unlock and image sensors ($p < .001$), and social media and phone unlock ($p < .05$). However there were no significant differences between phone unlock and finances ($p = .96$) and social media and image sensors ($p = .30$).

## Discussion

This paper develops a scale to measure the six perceptual characteristics outlined by DOI to assess people's perceived relative advantage, compatibility, complexity, trialability, observability, and reinvention that is adaptable across a variety of innovations from hardware-based to software-based innovations. Although DOI theory and its related constructs are well-established, researchers have struggled to consistently measure DOI and often fail to integrate reinvention. Previous scales measuring these characteristics either focused solely on one innovation, were not designed with all six perceptual characteristics in mind, or were not tested on software contexts. A strength of DOI theory lies in its ability to measure human perceptions of innovations across contexts; yet there has not yet been a scale developed for measuring DOI that is flexible to the wide range of innovations that we have seen and will continue to see as new technologies are created and old technologies are reformed.

Across three studies, we demonstrate that our six-factor structured questionnaire can be applied across different innovations with strong reliabilities (Cronbach's alpha > .75 for all six perceptual characteristics across all innovations). The questionnaire was effectively situated within a nomological network. The DOI scale predicted innovation use, was conceptually unrelated to algorithm awareness, and revealed that participants perceived each innovation differently. A final version of the DOI scale can be found in Table 17 and in S1 Table 1, with questions that are ready to be adapted to a variety of contexts, especially those that are software-based.

The final version of the scale contains three to four questions for each DOI characteristic. The questions demonstrate discriminant validity (meaning each factor is distinct from theoretically unrelated constructs) while also showing predictive validity by significantly predicting innovation use. Critically, the items are designed to allow researchers to apply questions across different types of innovations, including those that are software-based. Adapting the scale only requires the researcher to identify the innovation (e.g., facial recognition technology) and the task the innovation is designed to accomplish (e.g., unlocking my phone). Our study develops, tests, and adapts the scale using four types of image recognition software: facial recognition to unlock a phone, facial recognition to unlock a financial account, social media facial filters, and image sensors; however, the scale can also be adapted to other innovations (e.g., "[Facebook] allows me to accomplish tasks, such as [contacting my friends], more efficiently"). Other applications could consist of assessing the diffusion of new algorithms (e.g., predictive text algorithms) or emerging media (e.g., virtual environments). Our validated scale

**Table 17. Final Questions.**

| Relative Advantage | Compatibility | Complexity | Trialability | Observability | Reinvention |
|---|---|---|---|---|---|
| [Innovation] allows me to accomplish tasks such as [task] more efficiently | [Innovation] fits well with the way that I like to [task] | It is easy to get [innovation] to do what I want them to [task] | I have the ability to try out [innovation] to accomplish [task] before deciding whether I like it or not | I am able to observe when others in my environment use [innovation] to [task] | I often have to experiment with new ways of using [innovation] |
| [Innovation] is the best way to accomplish [task] | [Innovation] is completely compatible with my current way of [task] | Learning to operate [innovation] to accomplish [task] is easy for me | Trying out [innovation] to accomplish [task] has informed my decision to use [innovation] | My friends are able to observe the results of using [innovation] | I often have to modify [innovation] to get it to work for me |
| Using [innovation] helps me accomplish [task] better than not using [innovation] | [Innovation] suits my needs when [task] | My interactions with [innovation] is clear and understandable | I have had the opportunity to try [innovation] in the past | Others in my environment notice the impact of using [innovation] to [task] | I adapt [innovation] in a way that is different from how it was originally intended to be used |
| | [Innovation] integrates well with my current way of using [technology] | | | | |

ensures DOI theory remains methodologically equipped for age of artificial intelligence by providing researchers with a standardized tool to measure all six theoretical constructs across contexts.

This study is not without limitations. While the scale generally exhibited good internal discriminant validity, relative advantage and compatibility had a factor correlation greater than .85 across both samples in the finances and phone unlock innovation contexts. Within the context of diffusion of innovations, compatibility and relative advantage are the most similar perceptual innovation characteristics, so a high correlation is expected. Yet, they are conceptually distinct constructs and load well onto separate factors, therefore we have decided to keep them separate. Future studies should further investigate this by measuring an innovation where one would expect the compatibility of the innovation to be different from its perceived relative advantage. While our goal was to develop a DOI scale that is flexible to the range of innovations people use in their daily lives, we acknowledge that this scale may require further customization to be compatible with a given innovation.

Ultimately, this research develops a systematic approach for investigating the role of innovation characteristics in the diffusion of innovations that accommodates the study of hardware and software innovations and encompasses all six perceptual characteristics that contribute to innovation adoption (relative advantage, compatibility, complexity, trialability, observability, and reinvention). Across three studies, we constructed a validated and reliable scale to measure six DOI characteristics, with particular attention to the reinvention construct an often overlooked but conceptually critical component of diffusion theory. As algorithm-driven technologies such as artificial intelligence become pervasive, we anticipate that the capacity to modify or personalize these innovations will increasingly contribute to adoption decisions. By offering a generalizable tool grounded in psychometric evidence, this paper contributes both methodologically and theoretically to the study of how innovations are reshaped in practice. Future research can use this scale to investigate reinvention across diverse technologies, contexts, and populations.

## Supporting information

**S1 Table. Final Diffusion of Innovations Scale.** The final validated scale items for measuring all six DOI attributes. (DOCX)

**S2 File. Initial Items used to make adaptable diffusion of innovations scale.** The list of initial items used to measure all six DOI attributes across all contexts. (DOCX)

## Author contributions

**Conceptualization:** Hannah Overbye-Thompson.

**Data curation:** Hannah Overbye-Thompson.

**Formal analysis:** Hannah Overbye-Thompson.

**Funding acquisition:** Kristy A. Hamilton.

**Methodology:** Hannah Overbye-Thompson.

**Project administration:** Hannah Overbye-Thompson.

**Supervision:** Kristy A. Hamilton.

**Writing – original draft:** Hannah Overbye-Thompson.

**Writing – review & editing:** Hannah Overbye-Thompson, Kristy A. Hamilton.

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
