## [Decision Letter · Decision Letter 0]

20 Jun 2025

Dear Dr. Overbye-Thompson,

Thank you for submitting your manuscript to PLOS ONE. After careful consideration, we feel that it has merit but does not fully meet PLOS ONE’s publication criteria as it currently stands. Therefore, we invite you to submit a revised version of the manuscript that addresses the points raised during the review process.

**ACADEMIC EDITOR: **

Rajagopal, P. (2002). An innovation—diffusion view of implementation of enterprise resource planning (ERP) systems and development of a research model. *Information & Management* , *40* (2), 87-114.

Bradford, M., & Florin, J. (2003). Examining the role of innovation diffusion factors on the implementation success of enterprise resource planning systems. International journal of accounting information systems, 4(3), 205-225.

Jaradat, Z., Al-Dmour, A., Alshurafat, H., Al-Hazaima, H., & Al Shbail, M. O. (2024). Factors influencing business intelligence adoption: evidence from Jordan. *Journal of Decision Systems* , *33* (2), 242-262.

The paper could better position its contribution by addressing the reinvention construct. While most studies operationalize only Rogers’ five core attributes, reinvention is rarely integrated. This omission raises several questions which the paper can address:

Why is reinvention consistently excluded from DOI research, despite its theoretical relevance to post-adoption adaptation?

How does reinvention enhance the explanatory power of DOI in software innovation contexts, where iterative customization is common?

What methodological challenges hinder its operationalization, and how might this paper address them?

By engaging with these issues, the authors could strengthen their theoretical contribution and clarify how their scale advances the innovation literature (especially in software innovation). The following are comments from reviewers that also should be addressed.

We look forward to receiving your revised manuscript.

Kind regards,

Eu-Gene Siew

Academic Editor

PLOS ONE

Journal Requirements:

2. We noted in your submission details that a portion of your manuscript may have been presented or published elsewhere. [This is an original study reporting data that have not been reported beforehand, in addition to two datasets that were used in in our published study "Reinvention Mediates Impacts of Skin Tone Bias in Algorithms: Implications for Technology Diffusion" in the Journal of Computer Mediated Communication. While that paper focused on applying the scale to examine algorithm bias, the current manuscript provides the complete development and validation of the measurement instrument itself, offering a valuable methodological tool for scholars studying innovation diffusion.] Please clarify whether this [conference proceeding or publication] was peer-reviewed and formally published. If this work was previously peer-reviewed and published, in the cover letter please provide the reason that this work does not constitute dual publication and should be included in the current manuscript.

Reviewers' comments:

Reviewer's Responses to Questions

**Comments to the Author**

1. Is the manuscript technically sound, and do the data support the conclusions?

Reviewer #1: Partly

Reviewer #2: Yes

2. Has the statistical analysis been performed appropriately and rigorously?

Reviewer #1: Yes

Reviewer #2: Yes

3. Have the authors made all data underlying the findings in their manuscript fully available?

Reviewer #1: No

Reviewer #2: Yes

4. Is the manuscript presented in an intelligible fashion and written in standard English?

Reviewer #1: No

Reviewer #2: Yes

Reviewer #1: The initial and the final scale should be appended in the paper. It is very challenging for a reader to understand why the scale is good or not as they need to imagine how the scale looks like.

The paper is difficult to read as the text is full of phrases and numbers. Some of the numbers presented in the text can be put to a table to enhance readability.

The interconnection of the 3 studies should be explained as how they can contribute in achieving the goals of the paper.

The paper concludes with a discussion. It is important to provide concluding notes to alert the readers on the valuable contribution of the paper.

Reviewer #2: 1. The term "Nomological network" might not be understandable to general public. Please clarify what this network entails or why it matters could strengthen the impact.

2. "Creating an adaptable instrument for studying modern innovation diffusion" is promising but a little vague—perhaps specify what contexts or types of innovations it applies to.

3. The flow between the current challenges and the new research contribution could be smoother by adding more recent statistics.

4. The logic that if the constructs are "not positively associated," it "suggests an error in our measurements" is overly circular. It could also suggest that the theory may not fully apply in context of image recognition technology, or perhaps other variables are moderating the relationship.

5. The definition of the nomological network is too brief and generic — it doesn't clearly connect the definition to the specific research context. It would help to explain why testing relationships within a nomological network matters here,

or how this validates the scale.

6. "The questions are designed to discriminate among six DOI characteristics, while also predicting the frequency of innovation use". What "discriminate" means (discriminant validity? distinguish between constructs?).

7. "DOI theory will remain relevant...” is a generic claim with no connection to the specific contribution of this study.

8. This paper does not specify how this research moves the knowledge forward, or what makes it unique from past work. It says “this research is a starting point”, but it does not clearly articulate what the main contribution in creating a new way to operationalize DOI.

**Do you want your identity to be public for this peer review?** For information about this choice, including consent withdrawal, please see our Privacy Policy

Reviewer #1: No

Reviewer #2: No

---

## [Author Response · Author response to Decision Letter 1]

6 Aug 2025

A diffusion of innovations measurement scale for reinvention, relative advantage, compatibility, complexity, trialability and observability

PLOSOne EMID:a53c080254cdb6f1

AE.1 Comment

I have significant concerns about the positioning and claims made in this paper, particularly the assertion that 'Unlike most scales that seek to measure the perceptual characteristics outlined by diffusion of innovations, which mostly either focus on hardware innovations or do not include a measure of all six perceptual innovation characteristics.' This argument that the diffusion of innovations (DOI) theory is not focused on software innovations is poorly substantiated and lacks persuasive reasoning. For example, DOI has been explicitly applied to software-based innovations in prior research, such as enterprise resource planning (ERP) systems and business intelligence tools. See below (this is not an exhaustive list):

Rajagopal, P. (2002). An innovation—diffusion view of implementation of enterprise resource planning (ERP) systems and development of a research model. Information & Management, 40(2), 87-114.

Bradford, M., & Florin, J. (2003). Examining the role of innovation diffusion factors on the implementation success of enterprise resource planning systems. International journal of accounting information systems, 4(3), 205-225.

Jaradat, Z., Al-Dmour, A., Alshurafat, H., Al-Hazaima, H., & Al Shbail, M. O. (2024). Factors influencing business intelligence adoption: evidence from Jordan. Journal of Decision Systems, 33(2), 242-262.

The paper could better position its contribution by addressing the reinvention construct. While most studies operationalize only Rogers’ five core attributes, reinvention is rarely integrated. This omission raises several questions which the paper can address:

Why is reinvention consistently excluded from DOI research, despite its theoretical relevance to post-adoption adaptation?

How does reinvention enhance the explanatory power of DOI in software innovation contexts, where iterative customization is common?

What methodological challenges hinder its operationalization, and how might this paper address them?

By engaging with these issues, the authors could strengthen their theoretical contribution and clarify how their scale advances the innovation literature (especially in software innovation). The following are comments from reviewers that also should be addressed.

AE.1 Response

Thank you for your feedback. We have repositioned our contribution by addressing the reinvention construct. We have added a paragraph in the introduction on pages 3-4 to clarify that measurement scales within DOI research have overwhelmingly focused on Rogers’ five core attributes, often omitting reinvention, even in software-focused studies. We also added several paragraphs under “Reinvention” to substantiate on pages 8-11 our contribution to the reinvention construct. We report evidence of the ways that reinvention has been systematically excluded from empirical studies of DOI, and we define reinvention for its inclusion in a unified DOI construct by conceptualizing reinvention as a perceptual characteristic, rather than as an outcome or behavioral event. This resolves conceptual ambiguity and enables consistent measurement across innovations. We argue that this is especially relevant in software contexts, where iterative customization is common. Our scale captures this dimension, and we validate it across three diverse samples with strong psychometric evidence.

We believe the revised framing more accurately positions the paper’s contribution as addressing a longstanding measurement gap in DOI research, particularly around reinvention.

Journal Requirements:

JR.1 Comment

JR.1 Response

We have updated the manuscript to meet PLOS ONE’s style requirements, updating section headings, supplemental materials and table titles.

JR.2 Comment

2. We noted in your submission details that a portion of your manuscript may have been presented or published elsewhere. [This is an original study reporting data that have not been reported beforehand, in addition to two datasets that were used in in our published study "Reinvention Mediates Impacts of Skin Tone Bias in Algorithms: Implications for Technology Diffusion" in the Journal of Computer Mediated Communication. While that paper focused on applying the scale to examine algorithm bias, the current manuscript provides the complete development and validation of the measurement instrument itself, offering a valuable methodological tool for scholars studying innovation diffusion.] Please clarify whether this [conference proceeding or publication] was peer-reviewed and formally published. If this work was previously peer-reviewed and published, in the cover letter please provide the reason that this work does not constitute dual publication and should be included in the current manuscript.

JR.2 Response

Thank you for raising this point. The previously published article in the Journal of Computer-Mediated Communication (JCMC; "Reinvention Mediates Impacts of Skin Tone Bias in Algorithms: Implications for Technology Diffusion") was peer-reviewed and formally published. However, that article focused exclusively on a specific application of the DOI scale, namely, its use in analyzing perceptions of algorithmic fairness and skin tone bias.

In contrast, the current manuscript is an original contribution focused on the development and comprehensive psychometric validation of the DOI measurement instrument itself. This includes item generation and refinement, exploratory and confirmatory factor analysis, internal consistency, test–retest reliability, and measurement invariance testing, none of which were reported in the JCMC article.

Moreover, the present manuscript includes a dataset (Study 1) that was not included or described in the JCMC article. While two datasets are shared across both papers, they are used for entirely different analytic purposes: in the prior work, the scale was used as an independent variable to test a substantive theoretical model; here, the focus is on validating the scale as a methodological tool.

Therefore, this does not constitute dual publication because:

● The manuscripts address distinct research questions and serve different scientific purposes.

● There is no duplication of analyses or results.

● The current manuscript includes new data not previously reported.

● All overlapping data usage is transparently disclosed and appropriately cited.

JR.3 Comment

JR.3 Response

We have added a full ethics statement to the Methods section of the manuscript. The study protocol was reviewed and determined exempt by the University of California, Santa Barbara Human Subjects Committee (Protocol #4-23-0221) under Category 2 of 45 CFR 46.104(d). All participants across all study samples provided written informed consent prior to participation. No waiver of consent was granted or requested.

JR.4 Comment

JR.4 Response

We have added captions for the supporting information files at the end of our manuscript, and have updated the in-text citations accordingly.

Reviewers' comments:

Reviewer's Responses to Questions

Comments to the Author

RRQ.1. Is the manuscript technically sound, and do the data support the conclusions?

Reviewer #1: Partly

Reviewer #2: Yes

RRQ.2. Has the statistical analysis been performed appropriately and rigorously?

Reviewer #1: Yes

Reviewer #2: Yes

RRQ.3. Have the authors made all data underlying the findings in their manuscript fully available?

Reviewer #1: No

Reviewer #2: Yes

RRQ.3 Response

All data underlying the findings are available at Open Science Framework here under the files tab: https://osf.io/7p4gm/?view_only=2168ab9b71314b8eaf3d2bf254ec95ff

This link is also provided at the end of the manuscript.

RRQ.4. Is the manuscript presented in an intelligible fashion and written in standard English?

Reviewer #1: No

Reviewer #2: Yes

Reviewer #1:

R1.1 Comment

The initial and the final scale should be appended in the paper. It is very challenging for a reader to understand why the scale is good or not as they need to imagine how the scale looks like.

R1.1 Response

Thank you for your feedback. The initial scale used to make the paper is now appended as S2, with the final scale appearing both as an additional document (S1) as well as Table 17 at the end of the document.

R1.2 Comment

The paper is difficult to read as the text is full of phrases and numbers. Some of the numbers presented in the text can be put to a table to enhance readability.

R1.2 Response

Thank you for your feedback. We have reformatted some of the number-heavy paragraphs as tables, for example, on pages 16-17 and 20.

R1.3 Comment

The interconnection of the 3 studies should be explained as how they can contribute in achieving the goals of the paper.

R1.3 Response

Thank you for this suggestion. We’ve added a brief explanation on page 5 to clarify how the three studies work together. Following standard practices in scale development, Study 1 focused on item refinement, Study 2 validated the factor structure across domains, and Study 3 tested reliability and predictive validity. Together, they provide comprehensive evidence for the scale’s psychometric strength and utility.

R1.4 Comment

The paper concludes with a discussion. It is important to provide concluding notes to alert the readers on the valuable contribution of the paper.

R1.4 Response

Thank you for your feedback. We have added concluding remarks to the end of the discussion section to highlight the paper’s main contribution and its value for future research on innovation diffusion and technology design.

Reviewer #2:

R2.1 Comment

The term "Nomological network" might not be understandable to general public. Please clarify what this network entails or why it matters could strengthen the impact.

R2.1 Response

We have revised the manuscript to include a brief definition the first time the term is introduced, clarifying that it refers to the pattern of relationships between the scale and other psychological or behavioral constructs based on theory, which we hope should improve accessibility.

R2.2 Comment

"Creating an adaptable instrument for studying modern innovation diffusion" is promising but a little vague—perhaps specify what contexts or types of innovations it applies to.

R2.2 Response

Thank you for the suggestion. We have clarified in the manuscript that the instrument is designed to be adaptable across both software-based and hardware-based innovations, with particular relevance to contexts that permit reinvention. As algorithm-driven technologies such as artificial intelligence become increasingly pervasive, we anticipate that the capacity to modify or personalize these innovations will influence adoption decisions. To address this, we developed a construct that enables researchers to assess all six core characteristics of innovation as outlined in DOI theory. We validated the scale across four diverse application contexts—facial recognition for phone unlocking, social media filters, biometric security for financial transactions, and image-sensing faucets—capturing a broad spectrum of everyday software applications with varying perceptual and functional profiles.

R2.3 Comment

The flow between the current challenges and the new research contribution could be smoother by adding more recent statistics.

R2.3 Response

Thank you for this suggestion. We have revised the introduction to improve the flow between the field’s current measurement challenges and our contribution. Specifically, we have included challenges to why reinvention has not been consistently measured, and have included more recent references.

R2.4 Comment

The logic that if the constructs are "not positively associated," it "suggests an error in our measurements" is overly circular. It could also suggest that the theory may not fully apply in context of image recognition technology, or perhaps other variables are moderating the relationship.

R2.4 Response

Thank you for your feedback. We have revised our reasoning. You are correct that lack of association could indicate theoretical limitations or moderating factors rather than measurement error. We have reframed our hypothesis to acknowledge these alternative explanations while maintaining our theoretical expectation based on extensive prior DOI research.

R2.5 Comment.

The definition of the nomological network is too brief and generic — it doesn't clearly connect the definition to the specific research context. It would help to explain why testing relationships within a nomological network matters here,or how this validates the scale.

R2.5 Response

We have expanded our explanation of the nomological network concept to better connect it to our specific research context and explain its importance for scale validation. This revision clarifies how examining theoretically-predicted relationships validates our measurement approach.

R2.6 Comment.

R2 Comment 6. "The questions are designed to discriminate among six DOI characteristics, while also predicting the frequency of innovation use". What "discriminate" means (discriminant validity? distinguish between constructs?).

R2.6 Response

Thank you for requesting clarification on this important methodological point. We have revised this statement to clearly explain that "discriminate" refers to discriminant validity.

R2.7 Comment.

"DOI theory will remain relevant...” is a generic claim with no connection to the specific contribution of this study.

R2.7 Response

Thank you for your feedback. We have revised this conclusion to be more specific about how our scale contributes to DOI theory's continued relevance, connecting it directly to our research findings and the specific methodological advancement we provide.

R2.8 Comment.

This paper does not specify how this research moves the knowledge forward, or what makes it unique from past work. It says “this research i

---

## [Decision Letter · Decision Letter 1]

30 Sep 2025

A diffusion of innovations measurement scale for reinvention, relative advantage, compatibility, complexity, trialability and observability

PONE-D-25-06252R1

Dear Dr. Overbye-Thompson,

We’re pleased to inform you that your manuscript has been judged scientifically suitable for publication and will be formally accepted for publication once it meets all outstanding technical requirements.

Kind regards,

Eu-Gene Siew

Academic Editor

PLOS ONE

Additional Editor Comments (optional):

Reviewers' comments:

Reviewer's Responses to Questions

**Comments to the Author**

Reviewer #1: All comments have been addressed

2. Is the manuscript technically sound, and do the data support the conclusions?

Reviewer #1: Yes

3. Has the statistical analysis been performed appropriately and rigorously?

Reviewer #1: Yes

4. Have the authors made all data underlying the findings in their manuscript fully available?

Reviewer #1: Yes

5. Is the manuscript presented in an intelligible fashion and written in standard English?

Reviewer #1: Yes

Reviewer #1: (No Response)

**Do you want your identity to be public for this peer review?** For information about this choice, including consent withdrawal, please see our Privacy Policy

Reviewer #1: **Yes: ** Erniel Barrios

---

## [Editor Report · Acceptance letter]

PONE-D-25-06252R1

PLOS ONE

Dear Dr. Overbye-Thompson,

I'm pleased to inform you that your manuscript has been deemed suitable for publication in PLOS ONE. Congratulations! Your manuscript is now being handed over to our production team.

Kind regards,

on behalf of

Dr. Eu-Gene Siew

Academic Editor

PLOS ONE